# Effects of Exercise Training on Mitochondrial Fatty Acid β-Oxidation in the Kidneys of Dahl Salt-Sensitive Rats

**DOI:** 10.3390/ijms242115601

**Published:** 2023-10-26

**Authors:** Asako Namai-Takahashi, Junta Takahashi, Yoshiko Ogawa, Akihiro Sakuyama, Lusi Xu, Takahiro Miura, Masahiro Kohzuki, Osamu Ito

**Affiliations:** 1Division of General Medicine and Rehabilitation, Faculty of Medicine, Tohoku Medical Pharmaceutical University, Sendai 981-8558, Japan; 2Department of Internal Medicine and Rehabilitation Science, Graduate School of Medicine, Tohoku University, Sendai 980-8575, Japan

**Keywords:** chronic exercise, fatty acid β-oxidation, renal function, mitochondria, salt-sensitive hypertension

## Abstract

Exercise training (Ex) has anti-hypertensive and renal protective effects. In this study, we investigate the effects of Ex on mitochondrial fatty acid metabolism in the kidneys of Dahl salt-sensitive (Dahl-S) rats fed a high-salt (HS) diet. Eight-week-old, male Dahl-S rats were divided into three groups: (1) normal-salt diet, sedentary (NS-Sed), (2) HS diet, sedentary (HS-Sed), and (3) HS-Ex. The NS and HS groups were fed a diet containing 0.6% and 8% NaCl, respectively. The HS-Ex group performed treadmill running for 8 weeks (5 days/week; 60 min/day at 16–20 m/min, 0% gradient). Renal function and the expression of enzymes and regulators of β-oxidation and electron transport chain (ETC) complexes were assessed. HS increased systolic blood pressure and proteinuria, and Ex ameliorated these defects. HS also reduced creatinine clearance, and Ex ameliorated it. HS reduced the renal expression of enzymes of β-oxidation (carnitine palmitoyltransferase type I (CPTI) and acyl-CoA dehydrogenases (CADs)) and the related transcription factors peroxisome proliferator-activated receptor α (PPARα) and PPARγ-coactivator-1α (PGC-1α), and Ex restored this. HS also reduced the renal expression of enzymes in ETC complexes, and Ex restored this expression. Ex ameliorates HS-induced renal damage by upregulating enzymes involved in fatty acid β-oxidation and ETC complexes via increases in PPAR-α and PGC-1α expressions in the kidneys of Dahl-S rats. These results suggest that Ex may have beneficial effects on HS-induced mitochondrial dysfunction in the kidney.

## 1. Introduction

The kidney is considered to be one of the organs with the highest energy consumption levels [1,2,3,4]. Tubular epithelial cells have various functions, such as the regulation of fluid and electrolyte balance, the reabsorption of nutrients, and the excretion of circulating waste products [3,5]. These functions require a lot of energy, and mitochondria play an important role in energy production, through fatty acid metabolism and the electron transport chain (ETC). Peroxisome proliferator-activated receptor (PPAR)-γ coactivator-1α (PGC-1α) and PPAR-α regulate mitochondrial biogenesis and the expression of the enzymes that mediate fatty acid metabolism [6]. Medium-chain fatty acids freely diffuse into the mitochondria, and carnitine palmitoyltransferase type I (CPT-I) mediates the transport of long-chain fatty acids through the mitochondrial membrane, thereby initiating fatty acid β-oxidation. In mitochondria, acyl-CoA dehydrogenases catalyze the initial step in β-oxidation [7]. Acetyl-CoA, which is produced by β-oxidation, is the major substrate of the tricarboxylic acid (TCA) cycle and the ETC, and most of the ATP generated by aerobic respiration is produced through the ETC in a process referred to as oxidative phosphorylation [4].

Several previous studies have demonstrated that CPT-I, acyl-CoA dehydrogenase, PPAR-α, and PGC-1α play important roles in limiting acute kidney injury (AKI) and chronic kidney disease (CKD) [6]. The renal expressions of PGC-1α, CPT-I, and medium-chain acyl-CoA dehydrogenase (MCAD) are downregulated in mice with AKI or nephrosis [8,9]. Fibrates are PPARα agonists that regulate the expression of the genes encoding enzymes that mediate the β-oxidation of fatty acids. Clofibrate ameliorates the albuminuria, glomerulosclerosis, and renal interstitial injury of rats with a 5/6 nephrectomy [10], and fenofibrate inhibits the development of albuminuria and glomerulosclerosis through the upregulation of CPT-I and MCAD expression in mice fed a high-fat diet [11].

In PGC-1α knockout mice, renal dysfunction and tubular injury are exacerbated by ischemia-reperfusion injury [12]. However, tubule-specific PGC-1α transgenic mice show less severe AKI, better renal perfusion, quicker restoration of renal function, superior survival, and less severe histological injury [12]. Furthermore, PPAR-α knockout mice show a worse renal function and proximal convoluted tubule injury when subjected to sepsis-induced AKI [13]. Cpt1a-knockin reduces the expression of markers of fibrosis, the proinflammatory response, epithelial cell damage, and macrophage influx in models of renal fibrosis [14]. Finally, in patients with CKD, there is a positive correlation between the tubular expression of CPT-IA and estimated glomerular filtration rate, and the severity of fibrosis inversely correlates with CPT-IA expression [14].

We previously reported that exercise training (Ex) had renoprotective effects on several models of kidney disease. Ex on a treadmill ameliorated hypertension, proteinuria, glomerulosclerosis, and renal interstitial fibrosis in Wistar-Kyoto rats and spontaneously hypertensive rats (SHR) that underwent a 5/6 nephrectomy [15,16]. Furthermore, Ex attenuated the glomerulosclerosis and proteinuria of Goto-Kakizaki diabetic rats and Zucker diabetic rats without affecting their blood pressure levels [17,18]. In addition, Ex ameliorated the hypertension, proteinuria, glomerulosclerosis, and renal interstitial fibrosis of Sprague Dawley (SD) rats fed a high-fructose diet, and this was associated with higher renal expressions of PGC-1α, PPAR-α, and β-oxidation enzymes [19], suggesting that the renoprotective effects of Ex might have involved changes in mitochondrial fatty acid metabolism.

Dahl salt-sensitive (Dahl-S) rats were derived from SD rats by selective breeding to provide a tool to investigate high-salt (HS)-induced hypertension. HS intake induces hypertension and progressive cardiac and renal damage in Dahl-S rats [20,21,22,23], and they have much in common with patients with salt-sensitive hypertension, with respect to abnormalities in the nitric oxide (NO) system [24], insulin signaling [25], and lipid metabolism [25]. Previous studies have identified many pathophysiological mechanisms underpinning salt-sensitive hypertension, including abnormalities in the renin–angiotensin system, endothelin system, NO system, oxidative stress, the sympathetic nervous system, and 20-hydroxyeicosatetraenoic acid (20-HETE), which is a cytochrome P450 (CYP) 4A-derived metabolite of arachidonic acid [26,27]. HS intake also reduces the renal expressions of PPAR-α, MCAD, succinate dehydrogenase B, and cytochrome c oxidase [28], and fibrates attenuate the development of HS-induced hypertension and renal damage in Dahl-S rats [29].

Several previous studies have shown the beneficial effects of Ex in Dahl-S rats. Ex improves survival, ameliorates hypertension and chronic heart failure, improves endothelial function, and attenuates the muscle fiber-type shift of these rats [25,30,31,32,33]. We recently reported that Ex attenuated the HS-induced reduction in CYP4A expression and 20-HETE production in the kidneys of Dahl-S rats [34]. Because PPAR-α and PGC-1α are transcription factors that regulate the expressions of CYP4A and β-oxidation enzymes, the renoprotective effects of Ex may be mediated by the upregulation of not only CYP4A, but also of mitochondrial fatty acid β-oxidation, in Dahl-S rats. However, there are no studies regarding the relationship of Ex with renal fatty acid metabolism in Dahl-S rats. Therefore, in the present study, we aim to determine the effects of Ex on the expression of mitochondrial β-oxidation and ETC complex enzymes, PPAR-α, and PGC-1α in the kidneys of Dahl-S rats.

## 2. Results

### 2.1. Effects of HS and Ex on Plasma Parameters and Body Weight

Eight-week-old male Dahl-S rats were randomly assigned to a normal-salt (NS) diet (0.6% NaCl) diet + sedentary group (NS-Sed, *n* = 7), an HS diet (8% NaCl) + sedentary group (HS-Sed, *n* = 7), and an HS + exercise group (HS-Ex, *n* = 7). The HS-Ex group underwent 8 weeks of treadmill running.

One rat in the HS-Sed group died during the study and was excluded.

Table 1 shows the effects of HS and Ex on the plasma parameters and body weight at the end of the study. Blood samples were collected by decapitation on the final day of the experiment. The plasma albumin and glucose concentrations of the HS-Sed group were significantly lower than those of the NS-Sed group (*p* < 0.01 and *p* < 0.05, respectively), and the total cholesterol concentration was significantly higher (*p* < 0.01). The plasma glucose concentration was significantly higher in the HS-Ex group than in the HS-Sed group (*p* < 0.01), whereas the total cholesterol concentration was significantly lower (*p* < 0.01). The plasma free fatty acid concentration did not differ between the NS-Sed and HS-Sed groups, or between the HS-Sed and HS-Ex groups. There were no differences in the plasma total protein or triglyceride concentrations among the three groups. The body weight of the HS-Sed group was significantly lower than the NS-Sed group (*p* < 0.01), and there was no significant difference compared to the HS-Ex group.

### 2.2. Effects of HS and Ex on SBP, Urinary Protein, and Creatinine Clearance

In the final week of the study, the systolic blood pressure (SPB) was measured and the rats were placed in metabolic cages and their urine was collected.

After 8 weeks of the study, HS had significantly increased the SBP of the rats (*p* < 0.01), but Ex significantly ameliorated this effect (*p* < 0.01) (Figure 1a). HS significantly increased urinary protein exertion (*p* < 0.01) and Ex significantly reduced this effect (*p* < 0.05) (Figure 1b). HS also significantly reduced the creatinine clearance (*p* < 0.01) and Ex significantly ameliorated this effect (*p* < 0.05) (Figure 1c).

### 2.3. Effects of HS and Ex on the Expression of Acyl-CoA Dehydrogenase

A Western blot analysis was performed for the renal expressions of very long-chain, long-chain, medium-chain, and short-chain acyl-CoA dehydrogenase (VLCAD, LCAD, MCAD, and SCAD, respectively).

HS significantly reduced the expressions of LCAD, MCAD, and SCAD in the renal cortex of the rats (all *p* < 0.05), and Ex did not affect these expression levels (Figure 2b and Figure 3a,b). There were no differences in the VLCAD expression among the three groups (Figure 2a). HS significantly reduced the expressions of VLCAD, MCAD, and SCAD in the outer medullas of the rats (all *p* < 0.01), and Ex significantly ameliorated these effects (all *p* < 0.05) (Figure 2c and Figure 3c,d). There were no differences in the LCAD expressions of the three groups (Figure 2d).

### 2.4. Effects of HS and Ex on the Expressions of CPT-I, PPAR-α, and PGC-1α

A Western blot analysis was performed for the renal expressions of CPT-I, PPAR-α, and PGC-1α.

There were no differences in the CPT-I expressions of the three groups in the renal cortex (Figure 4a). HS significantly reduced the CPT-I expression in the outer medulla (*p* < 0.05) and Ex significantly ameliorated this effect (*p* < 0.05) (Figure 4d).

HS tended to reduce the PPAR-α expression in the renal cortex (*p* = 0.09), but Ex did not affect this (Figure 4b). There were no differences in the PGC-1α expressions among the groups (Figure 4c). HS significantly reduced the PPAR-α and PGC-1α expressions in the outer medulla (*p* < 0.01 and *p* < 0.05, respectively) and Ex significantly ameliorated these effects (*p* < 0.05 and *p* < 0.05) (Figure 4e,f).

### 2.5. Effects of Ex on the Expression of ETC Enzymes

A Western blot analysis was performed for renal expressions of complex I NADH ubiquinone oxidoreductase 1 β subcomplex 8 (CI-NDUFB8), complex II succinate dehydrogenase B (CII-SDHB), and complex IV mitochondrially encoded cytochrome c oxidase 1 (CIV-MTCO1).

HS significantly reduced CIV-MTCO1, CII-SDHB, and CI-NDUFB8 expressions in the renal cortex (*p* < 0.05, *p* < 0.01, and *p* < 0.01, respectively) (Figure 5a–c), and Ex significantly increased the CIV-MTCO1 and CI-NDUFB expressions (*p* < 0.05 and *p* < 0.05, respectively) (Figure 5a,c). Ex tended to ameliorate the effect of HS on CII-SDHB expression (*p* = 0.10) (Figure 5b). HS also significantly reduced CIV-MTCO1, CII-SDHB, and CI-NDUFB8 expressions in the outer medulla (all *p* < 0.05) (Figure 5d–f), and Ex tended to ameliorate the effect of HS on CIV-MTCO1 expression (*p* = 0.10) (Figure 5d).

## 3. Discussion

In the present study, we showed that Ex attenuated HS-induced hypertension and renal dysfunction in Dahl-S rats. HS reduced the renal expression of mitochondrial β-oxidation enzymes and the transcription factors that regulated their expressions, and Ex ameliorated these effects. Furthermore, HS reduced the renal expression of ETC complexes and Ex ameliorated this effect. These findings suggest that Ex might ameliorate HS-induced renal dysfunction by improving renal mitochondrial fatty acid β-oxidation and ETC function in Dahl-S rats.

Mitochondria are responsible for cellular energy production, and they are particularly abundant in organs that consume high levels of energy. The kidney is a mitochondria-rich organ, and previous studies have shown close associations between mitochondrial dysfunction and CKD progression [35]. Mitochondrial dysfunction is caused by inherited mitochondrial cytopathies and acquired defects, such as oxidative stress, hyperglycemia, dyslipidemia, proteinuria, and ischemia, and it causes podocyte injury, tubular epithelial cell damage, and endothelial dysfunction [35]. Some recent studies have shown abnormalities of the mitochondrial structure and function in the kidneys of Dahl-S rats. The activities of enzymes of the TCA cycle and mitochondrial ATP production were previously shown to be lower in the kidneys of Dahl-S rats than in the salt-resistant control strain SS.13^BN^, and HS consumption for 2 weeks further reduced ATP production [36]. In the medullary thick ascending limb (mTAL), which is responsible for NaCl reabsorption, there are fewer long mitochondria in Dahl-S rats fed an HS-diet for 7 days than in SS.13^BN^ or SD rats, which suggests that the ultrastructural abnormalities of mitochondria develop in the mTAL of Dahl-S rats prior to the development of histological signs of renal injury [37]. The expression of several mitochondrial proteins and oxygen utilization were lower in the mTAL of Dahl-S rats fed an HS-diet for 7 days than in SS.13^BN^ rats [38]. These findings suggest that mitochondrial dysfunction might be involved in the development of salt-sensitive hypertension. In addition, it has been reported that Ex reduces mitochondrial swelling and increases ATP formation in the kidneys of adult SHRs [39], suggesting that Ex might preserve mitochondrial function in the kidney.

The β-oxidation of fatty acids principally occurs in the mitochondria, and in the kidney, fatty acids are mainly taken up by proximal tubule cells [40]. Impairments in the oxidation of fatty acids are induced by stress stimuli, such as transient hypoxia, which causes renal injury through a decrease in ATP production, lipid accumulation, and an increase in fibrosis [40]. PPAR-α and PGC-1α are the key regulators of the expression of proteins involved in fatty acid uptake and oxidation, and CPT-I is considered to be the rate-limiting enzyme [41]. In patients with CKD, the renal expressions of CPT-I, PPAR-α, and PGC-1α are low and there is significant intracellular lipid accumulation [41]. Consistent with this, in mouse models of renal fibrosis, the renal mRNA expressions of PPAR-α and PGC-1α are lower and lipid accumulation is more marked [41]. In addition, HS consumption has been shown to reduce PPAR-α and MCAD expressions in the kidneys of Dahl-S rats [28] and to reduce the mRNA expression of mediators of fatty acid metabolism in the kidneys of SD rats [42]. We previously showed that the consumption of a high-fructose diet reduced the expressions of CAD and PPAR-α in the kidneys of SD rats and Ex increased the renal expressions of CADs, CPT-I, PPAR-α, and PGC-1α [19]. Interestingly, Ex increased the renal expressions of CADs, PPAR-α, and PGC-1α, even in SD rats fed a control diet, suggesting that Ex might have directly increases the renal expression of β-oxidation enzymes and the associated transcription factors [19]. The Ex-induced alterations in CAD and CPT-I expressions were found to be similar to those induced by fenofibrate [19]. These findings suggest that Ex might ameliorate HS-induced renal dysfunction by upregulating mitochondrial fatty acid β-oxidation and increasing PPAR-α and PGC-1α activities in the kidneys of Dahl-S rats.

Mitochondria are responsible for >90% of energy production in humans [35]. In aged Ames mice, the activities of the enzymes comprising kidney ETC complexes are significantly higher than those in wild-type mice, and it has been suggested that this superior ETC function contributes to their maintained renal function and longevity [43]. In the present study, we showed that Ex not only increased the expression of enzymes involved in mitochondrial fatty acid oxidation, but also that of ETC complexes in the kidneys of Dahl-S rats. Consistent with the present results, HS intake was shown to reduce the renal expressions of succinate dehydrogenase B and cytochrome c oxidases in Dahl-S rats [28]. Although HS consumption reduced the mRNA expression of TCA cycle enzymes, it did not substantially reduce the expression of enzymes in ETC complexes, and in fact increased the expression of complex V enzymes in SD rats [42]. In adult SHRs, Ex increased the activities of renal ETC complexes I and III, but did not affect those of complexes II and IV [39]. These findings are not consistent with the results of the present study, which suggests that the expression of ETC complexes might be influenced by whether the rat strain studied is susceptible to renal injury or not, the age of the rats, and the duration of the intervention.

Oxidative stress also plays a critical role in the progression of CKD [3,44]. Reactive oxygen species (ROS) have several sources, including mitochondrial enzymes, but the mitochondria themselves have an antioxidant system [35]. During mitochondrial respiration, the consumed oxygen is converted to superoxide radicals, which are further converted to hydrogen peroxide by manganese superoxide dismutase (SOD). Mitochondrial DNA is susceptible to ROS-induced stress and excessive ROS generation can lead to mitochondrial dysfunction. HS consumption has been shown to increase mitochondrial ROS production in the kidneys of Dahl-S rats [36], and we previously showed that HS increased nicotinamide adenine dinucleotide phosphate oxidase activity, xanthine oxidoreductase activity, and thiobarbituric acid reactive substances in the kidneys of Dahl-S rats, and that Ex ameliorated these effects [34]. Ex reduces mitochondrial ROS production in the kidneys of SHRs, and its renoprotective effects are similar to those of the mitochondria-targeting antioxidant MitQ_10_ [39]. Several previous studies have shown that PGC-1α and PPAR-α have antioxidant effects on the kidney. The PPARγ agonist rosiglitazone induces PCG-1α expression and increases SOD activity in the kidneys of db/db diabetic mice [45], and PGC-1α overexpression increases the number and respiratory capacity of mitochondria and protects renal tubular cells against oxidative stress in rabbits [46]. Furthermore, fenofibrate increases the expressions of antioxidant enzymes and CPT-I in the kidneys of mice fed a high-fat diet [11]. Thus, Ex-induced increases in PCG-1α and PPAR-α expressions might protect the mitochondria against oxidative stress in the kidneys of Dahl-S rats.

There were several limitations to the present study. First, we measured the expression of enzymes involved in β-oxidation and ETC complexes, and that of the transcription factors that regulated them, PPAR-α, and PGC-1α; however, we did not assess β-oxidation or ETC function themselves. Second, we did not assess the number or anatomical characteristics of the renal mitochondria. Third, we did not investigate the effect of blood pressure on the renoprotective effects of Ex. Previous studies have generated contradictory findings regarding the effects of Ex on blood pressure in Dahl-S rats. Although Ex significantly reduced their blood pressure in the present study, previous studies have shown that it ameliorated renal dysfunction independently of blood pressure [32,34]. Thus, further studies are needed to address these issues.

## 4. Materials and Methods

### 4.1. Animals

Male Dhal-S rats that were 6-week-old were obtained from Japan SLC Inc. (Shizuoka, Japan). The rats were housed in an animal care facility at a controlled temperature of 24 °C and humidity of 55% under a 12:12 h light–dark cycle with NS chow and water provided ad libitum. At the age of 8 weeks, the rats were randomly assigned to an NS-Sed group (*n* = 7), an HS-Sed group (*n* = 7), and an HS-Ex group (*n* = 7). The HS-Ex group underwent 8 weeks of treadmill running (KN-73 Tread-Mill; Natsume Industries Co., Tokyo, Japan) at 20 m/min at a 0° incline for 60 min per day for 5 days per week. The rats exercised for 10 min/day at an initial treadmill speed of 10 m/min at a 0° incline. The speed of the treadmill was increased gradually to 20 m/min and the duration of exercise training was increased to 60 min/day for 1 week. All of the procedures were conducted according to the guidelines of the Animal Welfare Committee of Tohoku University School of Medicine.

### 4.2. Measurement of Blood Pressure and Plasma and Urine Parameters

SBP was measured during the final week of the study using the tail cuff method (Coda Mouse & Rat Tail-Cuff Blood Pressure System, Kent Scientific Co., Torrington, CT, USA). The measurements were repeated more than 10 times until the rats became calm and stable SBP values were obtained.

The rats were placed in individual metabolic cages during the final week of the study, and 24 h urine samples were collected. The volume of each sample was measured, and then they were centrifuged at 1710× *g* for 15 min and the supernatants were collected and stored at −80 °C.

On the final day of the study, the rats were anesthetized with sodium pentobarbital (50 mg/kg, intraperitoneally), decapitated, and trunk blood samples were collected. These samples were centrifuged at 1710× *g* for 10 min and the supernatants were collected and stored at −80 °C. The plasma concentrations of total protein, albumin, glucose, triglycerides, free fatty acids, total cholesterol, and creatinine, and the urine concentrations of creatinine and protein, were measured using standard auto-analytical techniques (Nagahama LSL, Inc., Shiga, Japan).

### 4.3. Preparation of Kidney Samples

The kidneys of the rats were rapidly removed following decapitation and their masses were measured. The cortex and outer medulla of each left kidney were isolated carefully and stored at −80 °C. The frozen tissues were subsequently homogenized in 50 mmol/L of potassium phosphate buffer (pH 7.4) containing 1 mmol/L of ethylenediaminetetraacetic acid, 1 mmol/L of dithiothreitol, and 0.1 mmol/L of phenylmethylsulfonyl fluoride; the homogenates were centrifuged at 3000× *g* for 5 min; and the supernatants were collected and their protein concentrations measured using the Bradford assay [47].

### 4.4. Western Blot Analysis

The proteins in the samples of the kidney homogenate containing 30 μg of protein were separated by electrophoresis on a 10% sodium dodecyl sulphate polyacrylamide gel, then transferred electrophoretically to nitrocellulose membranes. The membranes were blocked by incubation in a TBST-20 buffer (10 mmol/L Tris-HCl, 150 mmol/L NaCl, 0.08% Tween 20, and 5% non-fat dry milk), then incubated at room temperature for 2 h with primary antibodies raised against CPT-I (Santa Cruz Biotechnology, Dallas, TX, USA), VLCAD, LCAD, MCAD, and SCAD (Santa Cruz Biotechnology), PPAR-α (Santa Cruz Biotechnology), PGC-1α (Santa Cruz Biotechnology), CI-NDUFB8 (Abcam, Cambridge, UK), CII-SDHB (Abcam), CIV-MTCO1 (Abcam), or β-actin (Sigma-Aldrich, St. Louis, MO, USA). The antibodies used were as follows: VLCAD, sc-376239, RRID: AB_10989696; LCAD, sc-82466, RRID: AB_2273323; MCAD, sc-49047, RRID: AB_2219689; SCAD, sc-107371, RRID: AB_2219814; PPAR-α, sc-9000, RRID: AB_2165737; PGC-1α, sc-13067, RRID: AB_2166218; CPT-I, sc-31128, RRID: AB_2229867; CI-NDUFB8, ab110413, RRID: AB_2629281; CII-SDHB, ab110413, RRID: AB_2629281; and CII-SDHB, ab110413, RRID: AB_2629281.

The membranes were washed with TBST-20 and then incubated at room temperature for 1 h with a horseradish peroxidase-conjugated anti-goat, anti-rabbit, or anti-mouse IgG secondary antibody (Santa Cruz Biotechnology). After further washing in TBST-20, specific protein bands were visualized using an enhanced chemiluminescent substrate (Super Signal; Thermo Fisher Scientific, Waltham, MA, USA).

The relative intensities of the bands were quantified using ImageJ ver. 2.0.0 (National Institutes of Health, Bethesda, MD, USA). The band intensities for each protein were normalized to those for β-actin as an internal standard, and the band intensity in the control NS-Sed group was assigned a value of 1.

### 4.5. Statistical Analysis

All the data are presented as the mean ± SEM. Comparisons of three groups were performed using one-way ANOVA followed by Tukey’s test for multiple comparisons. SPSS Statistics ver. 21.0 (IBM, Inc., Armonk, NY, USA) was used for this purpose. *p* < 0.05 was considered to represent a statistical significance.

## 5. Conclusions

Exercise training ameliorated high-salt-induced renal damage and upregulated the expression of enzymes involved in the β-oxidation of fatty acids and electron transport chain complexes in the kidneys of Dhal salt-sensitive rats. The renoprotective effects of exercise training may be mediated through improvements in mitochondrial function, mediated through increases in peroxisome proliferator-activated receptor α and peroxisome proliferator-activated receptor γ-coactivator-1α expressions.

## Figures and Tables

**Figure 1 ijms-24-15601-f001:**
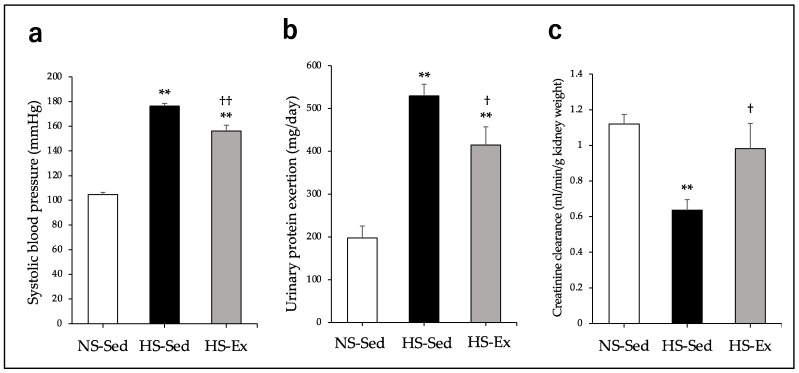
Impacts of high-salt intake and exercise training on systolic blood pressure, urinary protein exertion, and creatinine clearance in the last week were compared among the groups. (**a**) Systolic blood pressure; (**b**) urinary protein exertion; (**c**) creatinine clearance. NS-Sed, 0.6% normal-salt diet + sedentary group; HS-Sed, 8% high-salt diet + sedentary group; HS-Ex, high-salt diet + exercise group. Values are expressed as mean ± SEM. ** *p* < 0.01 vs. NS-Sed group, † *p* < 0.05, †† *p* < 0.01 vs. HS-Sed group.

**Figure 2 ijms-24-15601-f002:**
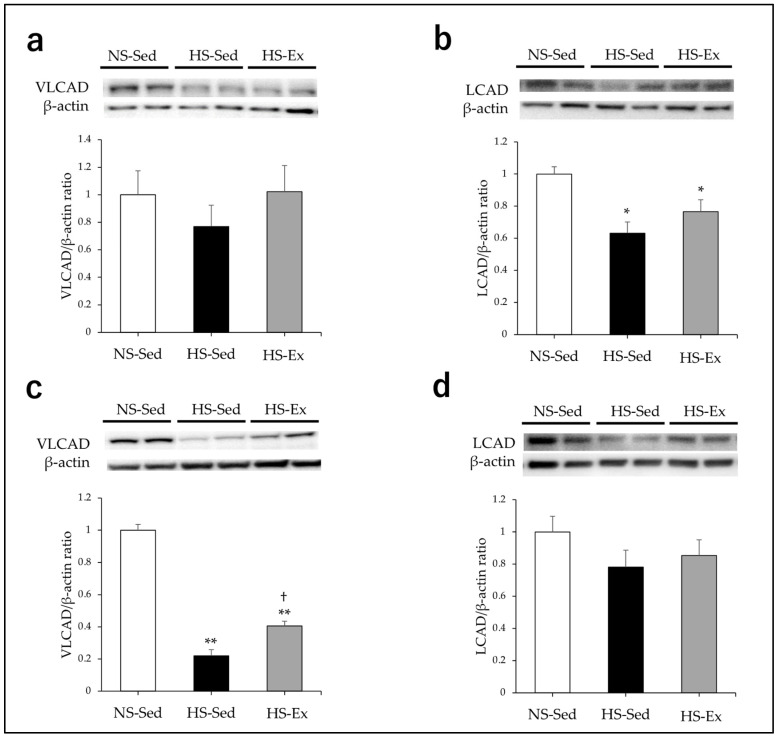
Impacts of high-salt intake and exercise training on renal expressions of VLCAD and LCAD. VLCAD and LCAD protein expressions were compared among the groups. (**a**,**b**) In the renal cortex; (**c**,**d**) in the outer medulla. NS-Sed, 0.6% normal-salt diet + sedentary group; HS-Sed, 8% high-salt diet + sedentary group; HS-Ex, high-salt diet + exercise group; VLCAD, very long-chain acyl-CoA dehydrogenase; LCAD, long-chain acyl-CoA dehydrogenase. Values are expressed as mean ± SEM. * *p* < 0.05, ** *p* < 0.01 vs. NS-Sed group, † *p* < 0.05 vs. HS-Sed group.

**Figure 3 ijms-24-15601-f003:**
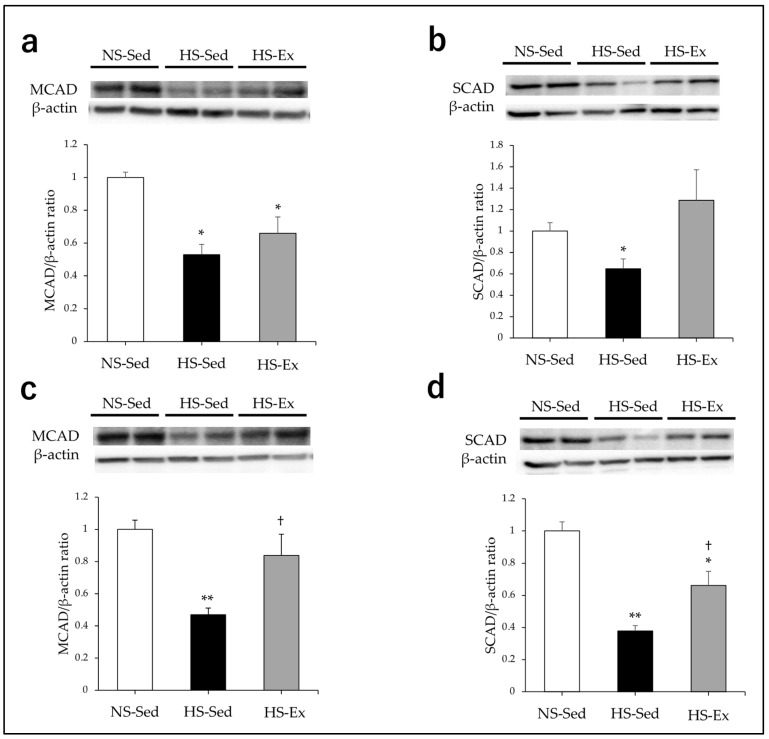
Impacts of high-salt intake and exercise training on renal expressions of MCAD and SCAD. MCAD and SCAD protein expressions were compared among the groups. (**a**,**b**) In the renal cortex; (**c**,**d**) in the outer medulla. NS-Sed, 0.6% normal-salt diet + sedentary group; HS-Sed, 8% high-salt diet + sedentary group; HS-Ex, high-salt diet + exercise group; MCAD, medium-chain acyl-CoA dehydrogenase; SCAD, short-chain acyl-CoA dehydrogenase. Values are expressed as mean ± SEM. * *p* < 0.05, ** *p* < 0.01 vs. NS-Sed group, † *p* < 0.05 vs. HS-Sed group.

**Figure 4 ijms-24-15601-f004:**
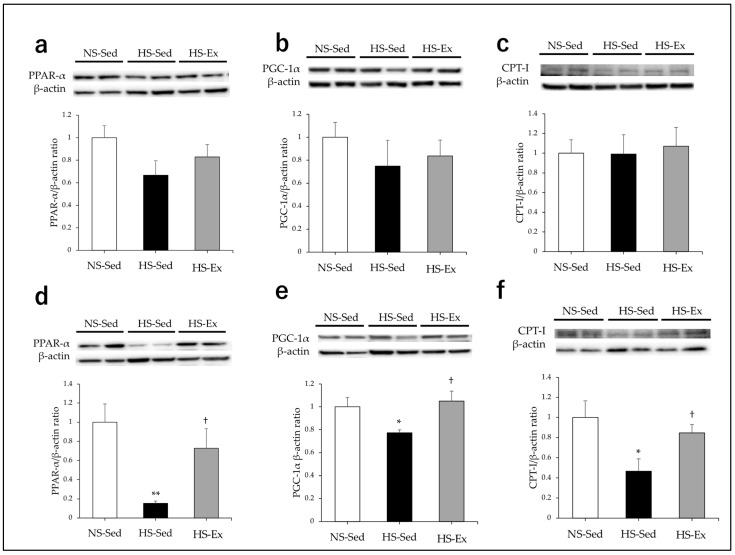
Impacts of high-salt intake and exercise training on renal expressions of CPT-I, PPAR-α, and PGC-1α. CPT-I, PPAR-α, and PGC-1α protein expressions were compared among the groups. (**a**–**c**) In the renal cortex; (**d**–**f**) in the outer medulla. NS-Sed, 0.6% normal-salt diet + sedentary group; HS-Sed, 8% high-salt diet + sedentary group; HS-Ex, high-salt diet + exercise group; CPT-I, carnitine palmitoyltransferase type I; PPARs, peroxisome proliferator-activated receptors; PGC, PPAR-γ coactivator. Values are expressed as mean ± SEM. * *p* < 0.05, ** *p* < 0.01 vs. NS-Sed group, † *p* < 0.05 vs. HS-Sed group.

**Figure 5 ijms-24-15601-f005:**
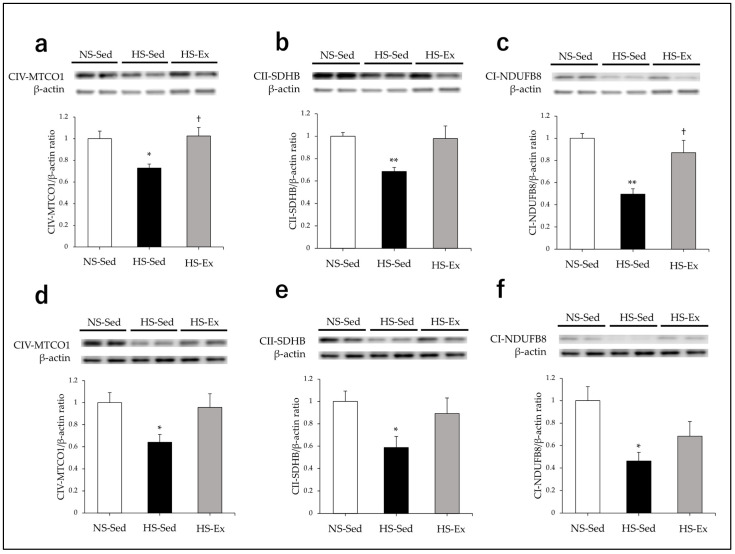
Impacts of high-salt intake and exercise training on renal expressions of CVI-MTCO1, CII-SDHB, and CI-NDUFB8. CVI-MTCO1, CII-SDHB, and CI-NDUFB8 protein expressions were compared among the groups. (**a**–**c**) In the renal cortex; (**d**–**f**) in the outer medulla. NS-Sed, 0.6% normal-salt diet + sedentary group; HS-Sed, 8% high-salt diet + sedentary group; HS-Ex, high-salt diet + exercise group; CVI-MTCO1, complex IV mitochondrially encoded cytochrome c oxidase 1; CII-SDHB, complex II succinate dehydrogenase B; CI-NDUFB8, complex I NADH ubiquinone oxidoreductase 1 β subcomplex 8. Values are expressed as mean ± SEM. * *p* < 0.05, ** *p* < 0.01 vs. NS-Sed group, † *p* < 0.05 vs. HS-Sed group.

**Table 1 ijms-24-15601-t001:** Effects of HS diet and Ex on plasma parameters and body weight.

	NS-Sed	HS-Sed	HS-Ex
Body Weight (g)	384.8 ± 5.3	331.0 ± 9.0 **	323.9 ± 10.3 **
Total protein (g/dL)	5.8 ± 0.1	5.7 ± 0.1	5.7 ± 0.1
Albumin (g/dL)	3.4 ± 0.1	2.8 ± 0.1 **	3.1 ± 0.2
Glucose (mg/dL)	156.9 ± 7.0	135.5 ± 2.4 *	155.4 ± 5.3 ††
Triglyceride (mg/dL)	111.0 ± 11.8	144.8 ± 16.3	117 ± 11.3
Free fatty acid (μEq/L)	141.9 ± 9.1	196.5 ± 26.6	202.0 ± 16.6 **
Total cholesterol (mg/dL)	61.0 ± 3.3	154.8 ± 5.2 **	112.7 ± 14.4 ** ††

NS-Sed, 0.6% normal salt diet + sedentary group; HS-Sed, 8% high salt diet + sedentary group; HS-Ex, high salt diet + exercise group. Values are expressed as means ± SEM. ** *p* < 0.01 vs. NS-Sed group, * *p* < 0.05 vs. NS-Sed group, †† *p* < 0.01 vs. HS-Sed group.

## Data Availability

The data presented in this study are available on request from the corresponding author. The data are not publicly available due to private reason.

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
