# Peer review of "Effects of Exercise Training on Mitochondrial Fatty Acid β-Oxidation in the Kidneys of Dahl Salt-Sensitive Rats"

_ijms, 2023, doi:10.3390/ijms242115601_

Round 1

Reviewer 1 Report

Effects of exercise training on mitochondrial fatty acid b-oxidation in the kidney of Dahl salt-sensitive rats                                                                       

The authors describe very well how aerobic exercise training can reduce the negative effects of a high-salt diet on kidney function. This is in line with a global increased salt intake and a more sedentary lifestyle.

The paper is well written, and the results are well described, however some minor issues must be adressed before publication.

Text is missing under results 2.1, and also the paragraph 2.2 is completely missing in the attached document. The first figure lacks number and text, but I guess this is an editorial issue. These sections have therefore note been evaluated.

Minor issues;

First of all, Table 1 is lacking a number and should also include body weight of the rats. Optimally also body composition fat mass/lean mass if available. A reduction in body weight can per se also introduce positive effects on plasma parameters and exercise performance.

The HS-diet and exercise started at the same time; it would be valuable to speculate if exercise could improve mitochondrial function in a similar way if exercise was introduced after a longer period on a HS-diet.

The treadmill running was a pre set protocol with no increase in speed or duration/ inclination over time. Most studies looking into aerobic exercise training increase the workload over time to make sure that a workload over 75% VO2max or heart rate max is kept. A short discussion on how different intesities of exercise will affect outcome should be included.

For all figures make sure font size is standardized throughout the paper.

Minor editing is needed.

Reviewer 2 Report

The manuscript entitled "Effects of exercise training on mitochondrial fatty acid beta-oxidation in the kidney of Dahl salt-sensitive rats" is an interesting study where the authors embarked on analyzing the impact of exercise on mito-FAO in the kidney. The model animal employed is Dahl salt-sensitive rat.

My specific comments are as under:

1. The grouping of animals seems limited. 3 groups were studied i.e, NS-Sed, HS-Sed, and HS-Ex. A fourth group with normal salt and exercise NS-Ex would have been very important for assessing the beneficial impact of exercise.

2. 8 week old were randomly selected. Were all of these rats sedentary after birth? 

3.  "The relative intensities of the bands in western blots were quantified by using ImageJ and the mean intensity of protein for NS-Sed group was assigned a value of 1". This is a poor description of the analysis of the western blot images, for example, in figure 4 c the weak intensity blots for CPT1  in HS-Sed and HS-Ex in comparison to strong intensity blot for b-actin in cortex resulted in a graph bar equal to 1, whereas the same blot intensity for PPAR-a and b-actin for HS-Sed in figure 4 d resulted in the graph bar equal to 0.2.

4. Figure 1 lacks the figure legend.

5. The table does not have a numerical number and the placement of the table is wrongly formatted as it has caused the text to disappear.

6. What is the probable cause or the reason for the increase in the plasma concentration of FFAs in both HS groups? Could the increase FAO might be because of these increased plasma levels of FAAs?

7. The last paragraph of the discussion highlighting the limitations of the current study should be removed.

8. The abbreviations should not be used in the conclusion section.

Round 2

Reviewer 2 Report

The authors have addressed all the points elegantly and improved the manuscript quality and context to the desired level. The manuscript should be accepted in its present form for publication.